# Multidimensional Modeling of the Number of Local Minimum-Energy Structures and the Energy of Putative Global Minima

Alejandro Varas,[*] Luciano Acevedo, Sebastián Carrasco, José Rogan, and Juan Alejandro Valdivia

*Departamento de Fisica, Facultad de Ciencias, Universidad de Chile*

E-mail: avaras@uchile.cl

**Abstract**

Using scarce-and-limited data about the number of local minimal energy structures and the energy of the global minima structures of the Lennard-Jones potential for $9 \leq N \leq 16$ atoms and dimension $1 \leq D \leq N - 1$, we propose new expressions to extrapolate these quantities. These expressions are obtained based on both multidimensional data fitting and neural networks to dimensionally move from known values, providing then new and more reliable extrapolations of the diversity of local minima energy structures, compared with previous methods based only on extrapolation of the number of three-dimensional local minimal energy structures for $N \leq 15$ atoms. Our estimations of the number of local minima in terms of $N$ and $D$ show that the maximum diversity of stable structures of the potential is found at $D = 3$. We also propose an expression to estimate the energies of the global minima energy structures, beyond the dimension $D = 3$ case. Surprisingly, we found that for each value of $N$, the energy of the global minimum can be divided into two regimes: when $D \leq \lceil N/2 \rceil$ and when $D \geq \lceil N/2 \rceil$, where $\lceil X \rceil$ means the whole part of $X$, each exhibiting a significantly

different trend from the other. While the data are limited, the observed trends appear robust, and the resulting fits provide reliable estimates even for values of $N$ beyond current computational feasibility.

# 1  Introduction

At the most basic of our perception, the space of our observable universe, is three dimentional. Despite this, because the space in which we live is three-dimensional, to understand the phenomena around us, we often try to reduce the dimensions of our space and model the space in two or even one dimension. By doing this dimensional reduction, phenomenology is simpler and therefore easier to assimilate. And once we have rationalized the fundamental concepts, we expand the dimensions of the model again in such a way as to recover our three-dimensional space[1,2].

However, there are remarkable examples of the use of additional dimensions to obtain solutions in our observable universe. Probably one of the most basic and succesful at the same time, is the set of complex numbers, which through the inclusion of a extra, imaginary dimension, allow us to rationalize an finally understood several mathematical problems that would be a dead end considering the single-dimension of the real numbers[3].

Of course, each time an increase in the dimensions of a model is introduced to understand certain phenomenology in a lower dimension, it can be counter-intuitive, especially if this increase in dimensionality means exploring beyond a third spatial dimension. The simple idea of a hyperspace of more than three dimensions requires some development of a capacity of abstraction and probably in some cases, an act of faith, so to speak. Why should a result in dimension $D+1$ be valid in dimension $D$? These types of questions are common in high-dimensional geometry and statistical physics, where structures and behaviors can simplify as the number of dimensions increases[4,5].

A simple way to begin to assimilate the idea that this may be possible is to accept that a line in one dimension will continue to be a line in two and three dimensions, therefore, it

would not be difficult to note that it will also continue to be valid in four or more dimensions. In the same way, a surface in a two-dimensional space will continue to be a flat surface in a three-dimensional space, therefore, it is likely that it will also continue to be a flat surface in four or more dimensions. The above leads us to the fact that any structure in a $D$-dimensional space will continue to preserve its $D$-dimensionality in a higher dimensional space.

An important problem to solve in our observable three-dimensional universe is to determine the geometric structure that atoms adopt to form nanostructures, since the vast majority of their properties depend on this geometric arrangement[1,2,6].

The Lennard-Jones potential (LJ) is a well known and also well established mathematical model, whose parameters can be fitted with experimental data to represent the forces to which any pair of atoms are subjected depending on the distance that separates them. Therefore, finding the stable atomic configurations of this potential is often a good approximation or at least a good starting point to determine the geometry and therefore the properties of matter at the nanoscale[7–9]

Remarkably, 2024 marks the 100th anniversary of the publication of the first paper by John Edward Lennard-Jones introducing this potential. This centennial has prompted a comprehensive reflection on its historical significance and enduring impact across physics, chemistry, and materials science[10]. Over the past century, the LJ potential has not only been a cornerstone in modeling intermolecular forces, but has also served as a testing ground for new theoretical and computational methods, ranging from phase behavior in condensed matter to the energy landscapes of atomic clusters.

The energy of the Lennard-Jones potential for a set of $N$ atoms is given by:

$$V_{LJ} = \sum_{i<j}^{N} 4\epsilon \left[ \left( \frac{\sigma}{r_{ij}} \right)^{12} - \left( \frac{\sigma}{r_{ij}} \right)^{6} \right],$$

where $r_{ij}$ is the distance between the $i$-th and $j$-th atom; $\sigma$ is the distance at which the potential is zero; and $\epsilon$ is the depth of the potential well. If we include different atom species,

then the $\sigma$ and $\epsilon$ parameters become species dependent.

Despite the apparent simplicity of the LJ potential, it has presented great challenges that have made it the subject of study over the years and still is today[10,11]. An example of this is the determination of the structure corresponding to the global minimum of energy as a function of the number of $N$ atoms. For certain values of $N$, finding these structures is a challenge and few methods can do it without relying on heuristics[12]. But even as $N$ increases, the certainty that the structure reported as the global minimum of energy decrease.

Another challenge offered by the Lennard-Jones potential, probably even more challenging than the previous one and certainly less studied, is the study of the diversity of the minimum energy structures as a function of the number of atoms $N$, *i.e.*, how many local minima exist for a given value of $N$? To date, there are only estimates that as $N$ increases, the number of local minima grows exponentially. Such estimates are based on the number of local minima found for $N \leq 15$, and of course, dimension $D = 3$[13,14].

Therefore, in this work we have taken up two challenges closely related to the previously stated ones. The first one, is to find an expression to estimate the number of local energy minima structures of the Lennard-Jones potential. It is important to highlight that we are not trying to find all the local minima structures, but to estimate how many there should exist. The second challenge, is to find an expression to estimate the energies of the global-minima structures of the Lennard-Jones potential. As well as in the first challenge, we remark, we are not trying to find the global-minima-energy structures, but to estimate their energies. As the data is scarce (limited to $D = 3$ in the case of the global minima energies, and relatively small $N \sim 14$ in the case of the number of local minima), our intention is to find expressions for a larger number of atoms, and not limited to dimension $D = 3$. Hence, to answer these challenges we have proposed expressions using a multidimensional perspective, based on the number of local-minima-energy structures, and the energies of the putative global-minima-energy structures of the Lennard-Jones potential for a few specific and limited values of $N$ and $D$. For a given value of $N$, calculations done in other dimensions $D \neq 3$ provide

additional information and restrictions that constrain the extrapolation formulas, not only for $D = 4$ but also for other smaller and larger dimensions.

Since previous predictions of the number of atoms of the LJ potential were built on the basis of the number of minima in $D = 3$ and relatively small values of $N$ (usually $N \leq 15$), it is expected that a functional approximation in terms of $N$ and $D$ will be more accurate than the previous ones.

In regard to the energies of global-minima structures, to the best of our knowledge, no estimations currently exist for these energies except for a very reduced number of cases. This gap in the literature is even more pronounced in cases where the spatial dimension satisfies $D \neq 3$.

## 2 Methods

Calculations of the number of local-minima energy structures and its energies were carried out using an extension from $D = 3$ to any value of $D$ of our minimization code[14], which has been successfully used in several of the authors' previous works[15–21]. In addition, we have also included the strategy developed by Locatelli *et al.*[22] to improve the probability of finding difficult Lennard-Jones clusters.

All local minima obtained were checked using the eigenvalues of the Hessian matrix, and to ensure that each local minimum considered is unique, *i.e.*, that for each pair of values of $N$ and $D$, we do not have repeated local minima, we use the ordered eigenvalues of the Coulomb Matrix[23].

At this point it is important to highlight our objective, which is to estimate the number of local minima of the Lennard-Jones potential as a function of the number of atoms, based on scarce-and-limited data on the number of local minima as a function not only of the number of atoms, but also of the dimension. Thus, our minimization code aims to find the greatest possible diversity of stable structures of the Lennard-Jones potential, including of

course the global minima for each $(N,D)$ pair, but in no case does it pretend to be a global minimizer. In terms of the number of minima for a certain pair of values of $N$ and $D$, the global minimum is just another local minimum, but that deserve particular attention as we will note below.

# 3 Results and Discussion

## 3.1 Number of local-minima-energy structures

We begin our data collection of the number of local-minima-energy structures in the Lennard Jones potential for different values of $N$, noting that, independent of the number of atoms $(N)$, in one dimension $(D = 1)$ there will only be one structure of minimum energy for each value of $N$, which will correspond to an unidimensional configuration of $N$ atoms.

On the other hand, let us note then that for $N = 2$, the minimum energy structure is unique and corresponds logically to a dimer, that is to say, a unidimensional configuration, regardless of the value of $D$. For $N = 3$, the minimum energy structure is also unique and corresponds to an equilateral triangle, $i.e.$ a flat two-dimensional structure, and this is so for every dimension $D \geq 2$. Then, for $N = 4$, the minimum energy structure is a regular tetrahedron, $i.e.$ a three-dimensional structure, and this is for every dimension $D \geq 3$, therefore, it is natural to assume that for any values of $N$ and $D$, such that $D \geq N-1$ there is only one minimum energy structure, corresponding to the so-called simplex-structure.

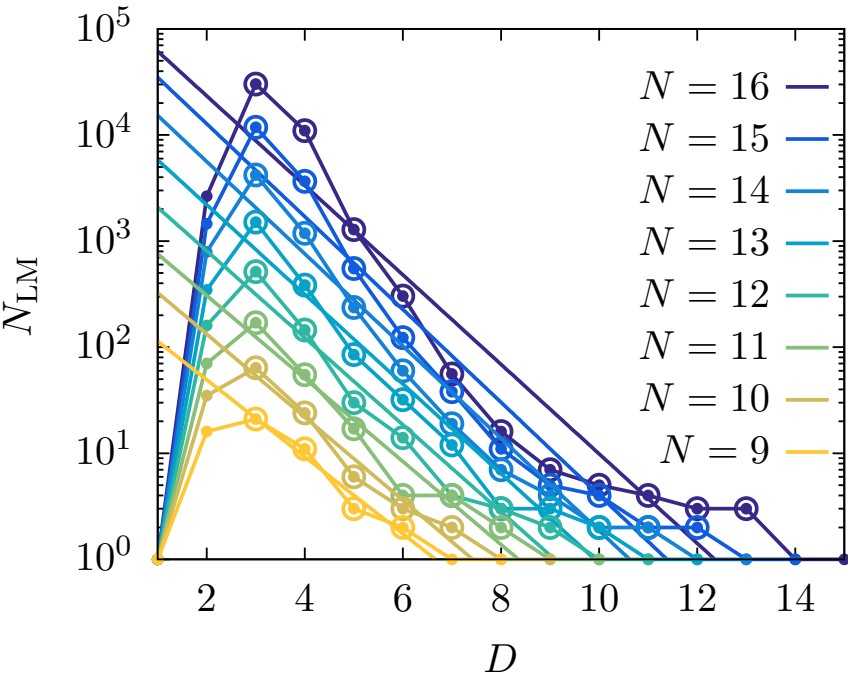

Figure 1: Number of local minima as a function of dimension $D$ for different values of $N$. The straight lines correspond to a fit using the data for $3 \leq D < N - 2$.

Figure 1 shows the number of local-minima energy structures $(N_{LM}(N,D))$ found for $9 \leq N \leq 16$ as a function of $D$, with $1 \leq D \leq N-1$. Of course, previous assumptions about the number of local minima energy structures for $D = 1$ and $D = N - 1$ were confirmed by calculations. Besides, it is surprising to observe how the number of local minima energy structures is always maximum for $D = 3$, which had already been observed by the pioneering work of Faken *et. al*[24], where they examined the role of dimensionality in the structure minimization problem with the Lennard-Jones potential. We will come back to this pont later.

The number of local minima for $1 < D < N-1$ represented on Fig. 1 should be regarded as lower bounds to the exact values, given the case that there might be local minima that we do not find with our method.

For the specific case $D = 3$, when $N \leq 13$ our method was able to find all the local minima reported in previous works[14]. We also found 4197, 11816, and 30215 local minima

for $N = 14$, $N = 15$, and $N = 16$ respectively.

A fit to the number of stable isomers for any $N$-atom cluster, with $9 < N < 16$, as a function of the dimension $D$, shows roughly an exponential growth as we reduce the dimension from $D = N - 2$ approximately, to $D = 3$, specially for the smaller values of $N$. At the same time, there is an exponential increase in the number of local minima as a function of $N$. Based on this observation, we first propose a rough estimation of the number of local-minima energy structures of the LJ potential given by:

$$N_{LM}(N, D) = e^{N-2.5-1.2D} \ . \tag{1}$$

It is interesting to note that equation 1 fits quite well for $D = 3$ with a previous estimation of the number of local minima of the LJ potential proposed by us ourselves[14]. In that work, we proposed $N_{LM} = e^{1.03N-6.13}$, and using $D = 3$ in equation 1 gives $N_{LM} = e^{N-6.1}$. However, it is evident from Figure 1 that the expression given by equation 1 is not very accurate, especially for larger values of $N$.

Furthermore, it should be noted that the fit given by equation 1 is only valid for $D \geq 3$. Therefore, it is valid to ask if it would be possible to obtain an expression to estimate the number of local minima energy structures being valid for any couple of values $N$, and $D$, that will also provide more accurate estimation for $D = 3$?

Considering the same points corresponding to the number of local minima as a function of the dimension $D$ for values of $N$ between 9 and 16, it is possible to provide a fit that considers all the values for $D$, finally putting additional restrictions to $D = 3$ and $D \neq 3$.

Examining Fig. 1, we note that the data presents a high concentration, and with a single maximum, in the number of local minima at low values of $D$ (around $D = 3$), and then decrease rapidly as $D$ increases. This indicates a long tail to the right, typical of positive asymmetric distributions (the number of local minima is always a positive quantity). This shape is consistent with the Log-Normal probability distribution, which we choose as an improved fit for the number of local minima.

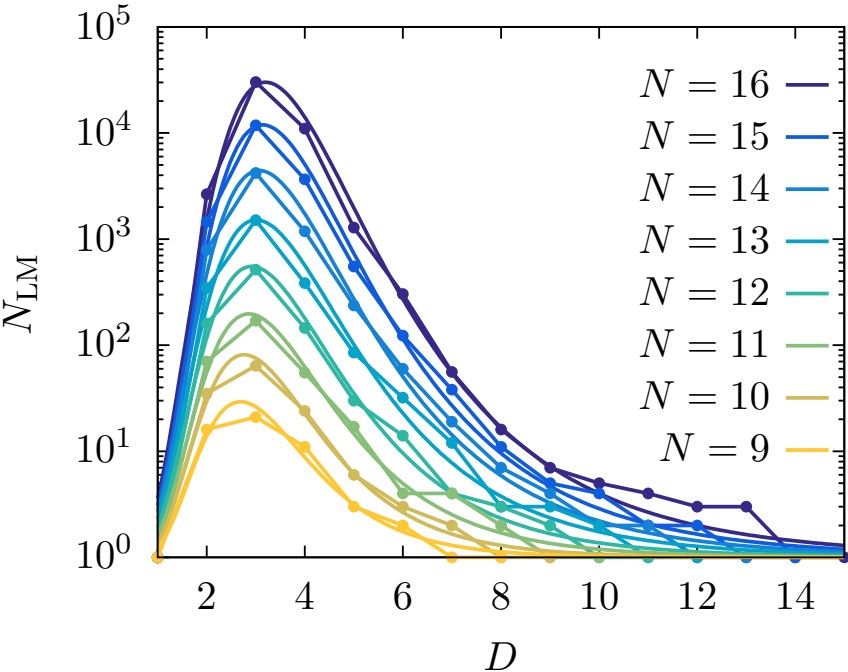

Figure 2: Dots shows the number of local minima as a function of dimension $D$ for different values of $N$. Lines shows a Log-Normal distribution fit of the data for each value of $N$.

Figure 2 shows the number of local minima for different values of $N$, and the corresponding fit using the logarithm of the data with a Log-Normal distribution given by:

$$\ln(N_{LM}) = \frac{A}{D\sigma\sqrt{2\pi}} \exp\left(\frac{-(\ln(D) - \mu)^2}{2\sigma^2}\right) , \tag{2}$$

where $\ln(x)$ is the natural logarithm of $x$, the values of $A$ (a constant to adjust the height of the distribution for each value of $N$), $\sigma$, and $\mu$ can be fitted for each value of $N$.

A quick look at Fig. 2 seems to show a good correspondence between the data and its fit. However, on closer inspection we notice some shortcomings. One shortcoming, is that as $N$ increases, the fit increases the number of local minima for $D = 1$. On the other hand, as $N$ increases, the fit does not accurately account for the number of local minima for $D$ close to $N - 2$.

To correct such shortcomings, we fit the logarithm of the data with a Log-Normal distribution multiplied by two aditional functions $f = f(D)$ and $g = g(D, N)$, thus:

$$\ln(N_{LM}) = f\,g\,\frac{A}{D\sigma\sqrt{2\pi}}\exp\left(\frac{-(\ln(D)-\mu)^2}{2\sigma^2}\right)\,, \tag{3}$$

where the values of $A$, $\sigma$, and $\mu$ can be fitted for each value of $N$. As we stated before, $f$ is a function to ensure that the number of local minima of the Lennad-Jones potential is 1 when $D = 1$, and $g$ is another function to ensure that the number of minima is 1 when $D = N - 1$, and $D = N - 2$. In particular we use:

$$f(D) = \tanh(D - 1)\,, \tag{4}$$

and

$$g(D, N) = \frac{1}{2}\left[\tanh(N - 2.5 - D) + \frac{1}{2}\right]\,. \tag{5}$$

Since the values of $A$, $\sigma$, and $\mu$ must be determined for each value of $N$, we look for functions $A = A(N)$, $\sigma = \sigma(N)$, and $\mu = \mu(N)$ such that they fit the data for $9 \leq N \leq 16$, which can then be extrapolated to larger values of $N$.

Table 1: Parameters $A$, $\mu$, and $\sigma$ for each value of $N$.

| $N$ | $A$ | $\mu$ | $\sigma$ |
|---|---|---|---|
| 9 | 13.730663 | 1.1314419 | 0.5577970 |
| 10 | 18.711303 | 1.1658799 | 0.5654135 |
| 11 | 25.205679 | 1.2320530 | 0.6115363 |
| 12 | 32.097339 | 1.2717576 | 0.6336778 |
| 13 | 40.415474 | 1.3330574 | 0.6622921 |
| 14 | 48.194069 | 1.3682530 | 0.6713059 |
| 15 | 55.479134 | 1.4030147 | 0.6724614 |
| 16 | 64.251931 | 1.4461522 | 0.6907929 |

Table 1 shows the values of $A$, $\sigma$, and $\mu$ for $9 \leq N \leq 16$. We tried and analized several

functions to fit data of Table 1, and the best results were obtained with:

$$A(N) = 0.22370N^2 + 1.74085N - 20.5972 \ , \tag{6}$$

$$\mu(N) = 0.04606N + 0.71849 \ , \tag{7}$$

$$\sigma(N) = 0.02004N + 0.38294 \ . \tag{8}$$

Figure 3 shows the data for $A$, $\sigma$, and $\mu$ together with their respective fits given by Eqs. (6-8). With these fits, Fig. 4 shows the surface that best fits the number of local minima of the Lennard-Jones potential as a function of the number of atoms $N$ and the dimension $D$ based on the Log-Normal distribution given by Eq. 3, with the dependency given by Eqs. (4-5), and the function fits provided by Eqs. (6-8). The result is now

$$\ln(N_{LM}) = f \, g \, \frac{A(N)}{D\sigma(N)\sqrt{2\pi}} \exp\left(\frac{-(\ln(D) - \mu(N))^2}{2\sigma(N)^2}\right) \ ,$$

It is worth emphasizing the strengths of the proposed fit, some of which have been already stated above. In particular, the fit given by Eq. (3) can estimate accurately the number of local minima of the Lennard-Jones potential for $D = 1$ and $D = N - 1$; and more precisely than previous expressions for $D$ around $N - 2$. Moreover, our model provides a reasonable estimate for the number of local minima of the Lennard-Jones potential in three dimensions, particularly for values of $N$ close to those used to build the model ($9 \leq N \leq 16$). Within this range, the fit accurately captures the log-normal distribution of minima across dimensions. However, as $N$ increases beyond the training window, it is natural to expect a reduction in precision, since the extrapolation must rely on trends inferred from significantly smaller systems. Table 2 shows some extrapolation estimates about the number of local minima with the fit used in the present manuscript and comparison with previous works. As stated before, the discrepancies between the predicted number of local minima from our model and the previously reported values in Table 2 can be attributed to several factors. First,

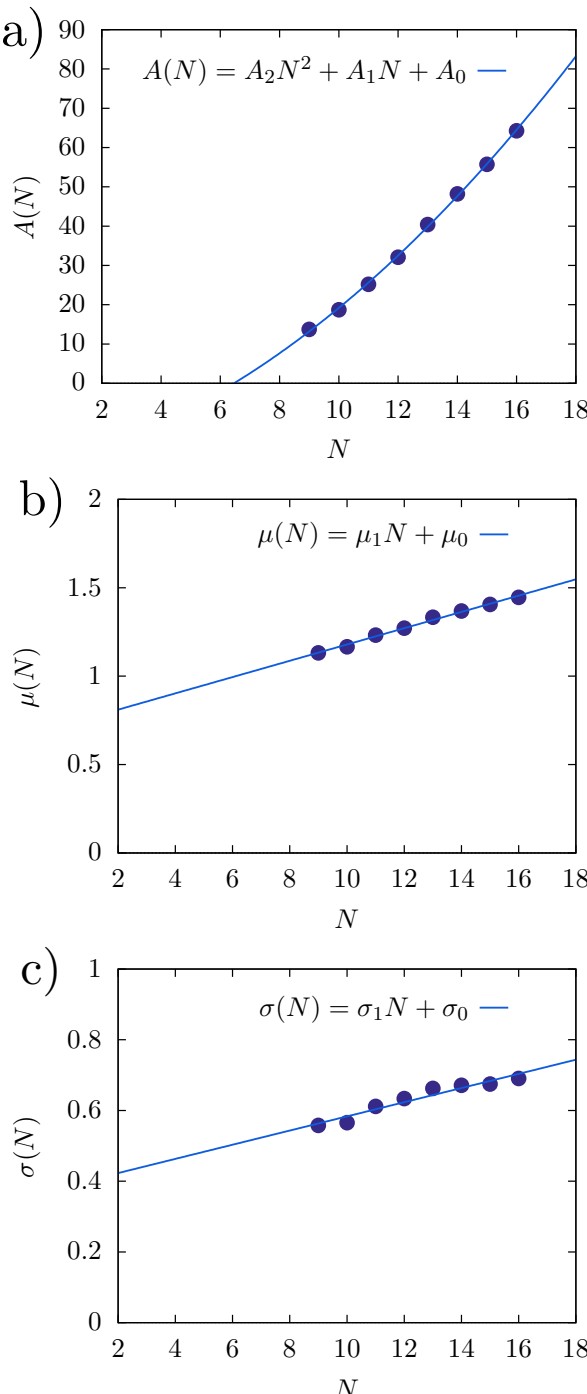

Figure 3: Panels a), b), and c) show the values of $A$, $\mu$, and $\sigma$, respectively, along with their corresponding fits.

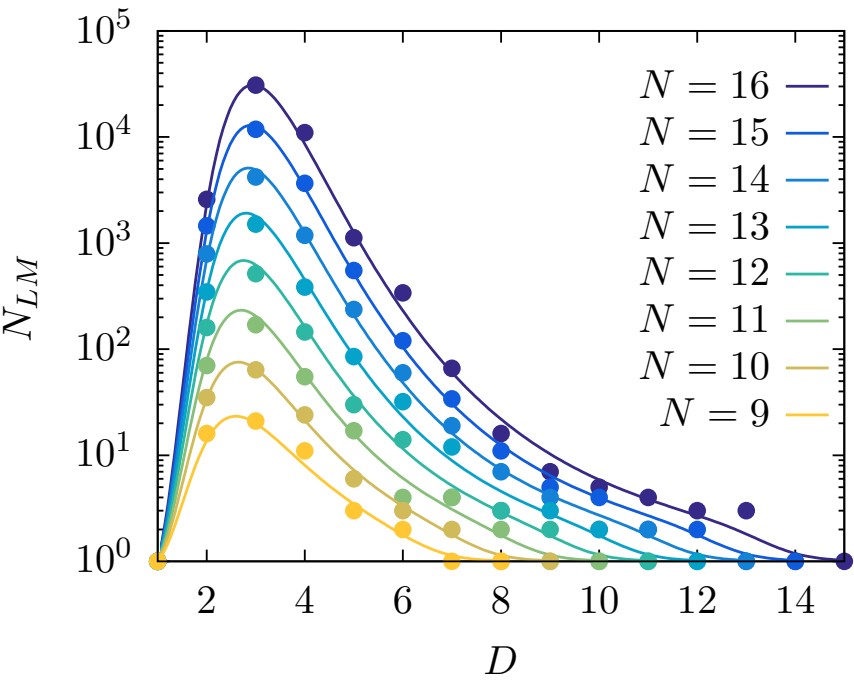

Figure 4: Number of local minima as a function of dimensionn $D$, and number of atoms $N$.

the extrapolation is based on data for relatively small systems, specifically in the range $9 \leq N \leq 16$, so it is expected that the accuracy of the model decreases as we move further from this training range. However, it is important to note that as we considered other dimensions beyond $D = 3$, we have added a larger number of restrictions into the system, that is expected to provide a better extrapolation to the number of minima in $D = 3$ and other dimensions as we increase $N$. Second, historical estimates of the number of minima for $N = 147$ have shown extreme variability: notably, between 1993 and 1997, published predictions decreased by almost 200 orders of magnitude. This dramatic shift underscores the sensitivity and uncertainty inherent in estimating the complexity of high-dimensional energy landscapes.

Third, although we have not placed any restrictions on the setting so that the number of local minima of the Lennard-Jones potential has a maximum at $D = 3$ for all $N \geq 9$, our fit naturally accounts for it, even for large values of $N$, far beyond the current computational capabilities and energy requirements needed to obtain all the local minima structures for

Table 2: Estimated number of local minima for the Lenard-Jones potential for some selected values of $N$.

| $N$ | Reported in this work | Previously reported |
|---|---|---|
| 38 | $\sim 5.7 \times 10^{8}$ | $\sim 2.5 \times 10^{14}$ [14] |
| 55 | $\sim 7.1 \times 10^{9}$ | $\sim 10^{10}$ [25] and $\sim 10^{22}$ [26] |
| 147 | $\sim 2.6 \times 10^{13}$ | $\sim 10^{60}$ [26] and $\sim 10^{259}$ [27] |

such values of $N$. Figure 5 shows a heat map, normalized with the maximum number of local minima provided by our fit for each value of $N$.

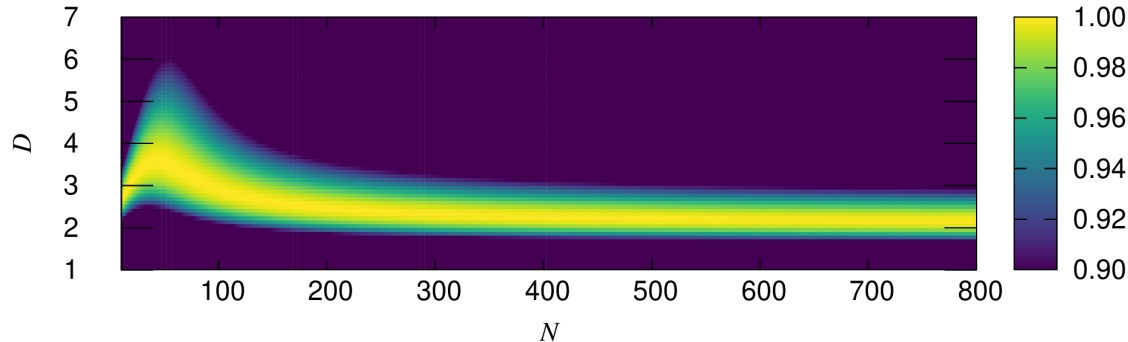

Figure 5: Heat map of the number of local minima predicted by the fit as a function of dimension $D$ and number of atoms $N$, normalized by the maximum value for each $N$. The yellow line highlights, for each $N$, the dimension $D$ at which the diversity (i.e., number of local minima) reaches its maximum.

The yellow color in Fig. 5 corresponds to the areas where the diversity is maximum for each value of $N$. It is observed that for small values of $N \leq 100$, the maximum diversity according to our fit is found approximately for $3 \leq D \leq 4$. On the other hand, for large values of $N$, the maximum diversity tends to stabilize approximately for $2 \leq D \leq 3$. Here, it is important to keep in mind that the heat map corresponds to our fit, which is a continuous function of $N$ and $D$, however, the number of local minima as a function of $D$ and $N$ is a discrete function, therefore, although the maximum diversity of our fit could be found in different ranges depending on the $N$, it is possible that the maximum diversity will be always in dimension $D = 3$. This last point is something that to the best of our knowledge has not

been fully explored. Faken *et al.*[24] observed that at least up to $N = 15$, this was a feature of the Lennard-Jones potential; however, they did not explore why this phenomenology occurred.

Let us start with the calculated data we have, *i.e.*, for small $N$ values, from $D = 1$, where there is only one local minimum, the maximum diversity of local minima energy structures is found at dimension $D = 3$ and decreases as the dimension increases, until in dimension $D = N - 2$ and $D = N - 1$ there is only one minimum.

When we consider a pair potential such as the Lennard-Jones potential, the total potential energy of the system depends on the relative positions of the atoms, and the local minima of this energy correspond to stable or metastable configurations where the system can reside. In dimension one ($D = 1$), the atoms are restricted to one line. Due to the nature of the pair potential and spatial constraints, there is only one configuration in which the atoms can be arranged to minimize the energy: a linear chain with optimal spacing between atoms. Any perturbation to this arrangement increases the energy, and there is not enough space for the atoms to rearrange differently without increasing the energy. Therefore, there is only one local minimum.

By increasing the dimension to $D = 2$, the atoms have more freedom to rearrange themselves in the plane. This allows for a greater variety of stable configurations, such as triangular, hexagonal, or similar arrangements, which increases the number of local minima. In three dimensions ($D = 3$), the spatial freedom is even greater. Atoms can form more complex structures, as well as dense and generally symmetric packing such as three-dimensional icosahedral or decahedral clusters. This significantly increases the number of ways in which atoms can organize themselves to reach configurations of minimum energy, resulting in a maximum in the number of local minima compared to other dimensions. As the dimension continues to increase beyond $D = 3$, although the atoms have more degrees of freedom, an interesting phenomenon occurs. In higher dimensions, the available space grows so rapidly that atoms can more easily "avoid" each other. This means that atoms are less likely to in-

teract significantly with many close neighbors, as the effective density of neighbors decreases in high-dimensional spaces. As a result, the potential energy landscape becomes smoother and features fewer local minima, since interactions that could create local energy wells are less frequent. In fact, due to this effect, by the time we reach the dimension $D = N - 2$ and $D = N - 1$, there is only one minima.

The latter is what is known as the *Concentration of Measure Phenomenon*[4], a fundamental property in high-dimensional spaces that describes how, as the dimension of the space increases, most of the volume or size of that space is concentrated around a specific region or mean values. In simple terms, in high-dimensional spaces, "almost all" points are "close" to each other in some probabilistic sense, and the functions defined in these spaces tend to have values very close to their mean over most of the domain. This implies that the configurations of atoms tend to be less varied in terms of potential energy, as the distances between pairs of atoms become more uniform. As a result, the potential energy landscape is simpler and has fewer local minima.

Another mathematical argument supporting what has just been discussed is the *Hard Sphere Packing Factor*[6]. At $D = 3$, the packing of hard spheres has a known maximum density (approximately 74%), which allows for multiple dense arrays; however, in higher dimensions, the maximum packing density decreases drastically. For example, at $D = 4$, it is about 45%. Lower packing density implies fewer ways to arrange atoms in dense and energetically favorable configurations.

Finally, in dimension $D = N - 1$, where $N$ is the number of atoms, the spacing is so large compared to the number of particles that the atoms can be distributed in a way that minimizes the energy without significant interaction constraints. In this case, similar to $D = 1$, there is only a local minimum because there is enough space for the atoms to be placed in positions that minimize repulsive interactions without the need for multiple distinct configurations.

In summary, from our point of view, for low values of $N$, the number of local minima

reaches a maximum at $D = 3$ due to an optimal balance between degrees of freedom and significant interactions between atoms. In lower dimensions, the spatial constraint limits the possible configurations, while in higher dimensions, the excess space reduces the probability of interactions reducing the possibility of establishing multiple local minima.

At this point we might ask, whether for higher values of $N$, where due to computational limitations we cannot exhaustively explore the energy landscape, the maximum diversity of the local Lennard-Jones minima remains at $D = 3$ or shifts to another dimension, lower or higher. Unfortunately, with the local minima diversity data up to $N \leq 16$, $D \leq N - 1$ and with the proposed fit, we cannot yet answer the latter question with certainty; however we can try to explain what would be the reasons for the maximum diversity to remain at $D = 3$ or to shift to another dimension.

In two dimensions $(D = 2)$, as $N$ increases, the atoms may face tighter spatial constraints by being confined to a plane. This may generate geometrical frustration, where it is difficult for all particles to reach optimal interactions simultaneously, which could lead to an increase in the number of local minima, but with high values of their energy. Between these local minima, where it is difficult for the vast majority of atoms to reach optimal interactions simultaneously, it could be possible to find numerous structures with defects such as dislocations and vacancies that may be more stable, contributing to a greater variety of minimum energy structures.

On the other hand, in four dimensions atoms have an additional degree of freedom to reorganize themselves, which may reduce the geometric frustration present in lower dimensions. As the number of atoms increases, the system's ability to accommodate local optimal arrangements improves in higher dimensions, allowing more atoms to simultaneously experience favorable interactions. This effect, however, introduces a trade-off: when many atoms are comfortably arranged, the system becomes less frustrated and may present fewer competing configurations, thus potentially reducing the number of distinct local minima. In contrast, when geometric constraints are stronger—as in lower dimensions—a greater number

of atoms remain frustrated, which can lead to a richer variety of metastable arrangements. Therefore, the observed diversity is the result of a balance between the flexibility provided by additional dimensions and the level of frustration that allows for multiple distinct minima. In particular, three dimensions appear to offer the right balance between these competing effects, leading to the maximum in diversity observed for $D = 3$.

We may wonder why the above multidimensional discussion is import. As we observe the variation of the number of local minima with respect to the dimension $D$ for fixed value of $N$ in Fig. 1, we notice a distinct pattern of $N_{LM}$ with $D$ that seems to approximately repeat as we vary $N$. Consequently, it is worth evaluating if one could use that pattern to find the number of local minima $N_{LM}$ starting from know values at large $D$ and try to predict the values in lower dimensions. The reason is that local minima at higher dimensions ($N \gg 3$) are generally less costly to find, as mentioned throughout the manuscript. We train a neural network to perform this task to offer a proof-of-principle of the idea. For the sake of the example, we can think of it as a function

$$\mathcal{N} = \mathcal{N}\left[D, N_{LM}(D+1, N), N_{LM}(D+2, N)\right]. \tag{9}$$

The goal is to adjust the neural net parameters, or weights, to make it *learn* how to extrapolate the number of minima at dimension $D$ given the numbers of minima at dimensions $D+1$ and $D+2$. In practice, by iterating this process, the output should start to approximate $N_{LM}(D, N)$. Consequently, we must minimize the mean squared error (MSE)

$$\text{MSE} = \langle\{N_{LM}(D, N)$$
$$- \mathcal{N}\left[D, N_{LM}(D+1, N), N_{LM}(D+2, N)\right]\}^2\rangle,$$

which is the figure-of-merit that we use for training. It is worth noting that the neural

net is unaware of the number of atoms by design. This is explicitly done to force it to learn the general pattern.

In Fig. 6, we show a comparison between the iterated prediction that the neural net offers for $N = 16$, starting with the calculated points for $D \geq 6$ and $D \geq 7$, which are relatively easy to find, *i.e.*, there is a factor of 100 and 1000 fewer minima compared with $D = 3$, respectively. The neural net is trained to minimize the MSE using data from $N = 9$ to $N = 15$ for different values of $D$. Thus, it is not trained with $N = 16$ data. Nevertheless, it offers an excellent agreement when the iterated prediction is initialized with the numbers of minima for $D = 6$ and $D = 7$. This illustrates further that less costly higher-dimension calculations can provide insight into low-dimensional ($D = 3$) results.

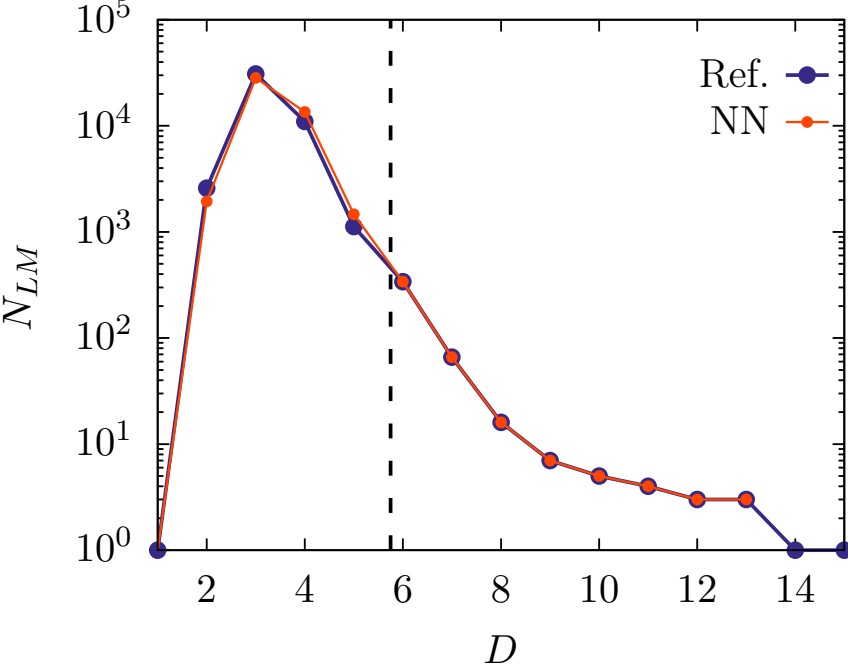

Figure 6: Comparison between iterated prediction that the neural net gives with the computed number of local minima for $N = 16$. We use the known data at the right of the dashed vertical line to estimate the first point at the left. Then, the prediction is fed into the neural net to generate the next prediction and repeat.

## 3.2 Energies of global-minima structures

Despite the great efforts and different minimization techniques applied to the Lennard-Jones potential in order to obtain the structures corresponding to the global minima in dimension $D = 3$, the energies of these stable structures have been little explored in terms of their dependence with dimension $D$. And, as we it occurred with the number of local minima, a multidimensional analysis can provide additional useful information and restrictions about its dependence with $N$ for $D = 3$ and $D \neq 3$. Figure 7 shows the energies of the global minima for $9 \leq N \leq 16$ as a function of the dimension $D$ that we have found with our method for $9 \leq N \leq 16$.

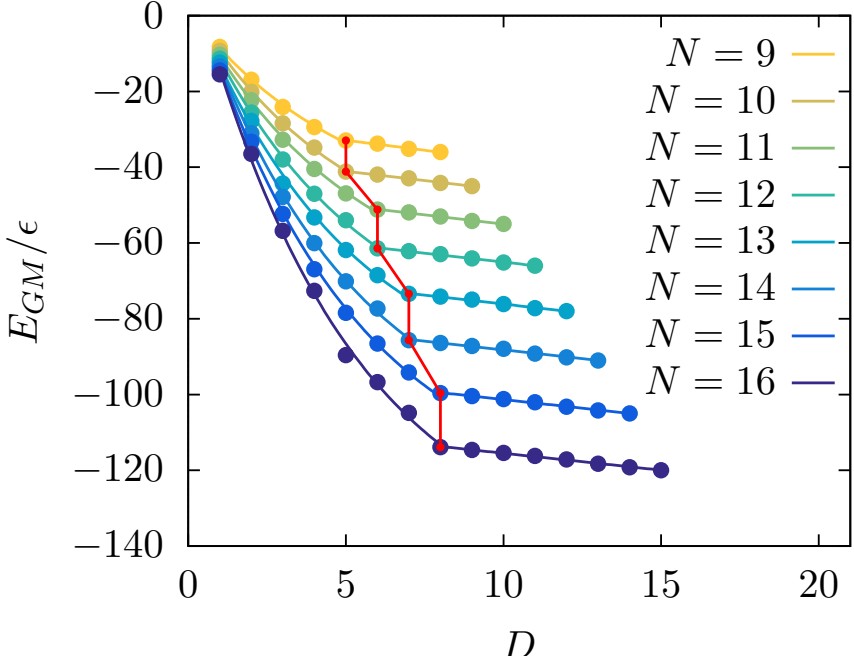

Figure 7: Energies of putative global minima as a function of dimension $D$, and number of atoms $N$.

Although for all combinations of $N$ and $D$ we are not sure that there is no other structure with lower energy, we have some certainties for some combinations of $N$ and $D$. As we have said before, in dimension $D = 1$ there is only one structure of minimum energy, therefore, since our method is able to find a local minimum for every value of $N$ in dimension $D = 1$,

we are sure that its energy corresponds to that of a global minimum. The same is true in dimension $D = N - 1$. Finally, we have compared and made sure that all energies of the global minima found in dimension $D = 3$ are equal to those published in the Cambridge Cluster Database[28].

By analyzing the data plotted in Fig. 7, we observe that the energy of the global minima for each value of $N$ decreases significantly with increasing dimension $D$, but this decrease slows down in higher dimensions. Specifically, from $D = 1$ to $D = (N + 1)/2$ if $N$ is odd, or from $D = 1$ to $D = N/2$ if $N$ is even, the energy decreases rapidly. However, from $D = (N + 1)/2$ for odd $N$ values and from $D = N/2$ for even $N$ values, the reduction in energy becomes much more gradual, approaching a limiting value at $D = N - 1$ which corresponds to $E_{GM}(N, D = N - 1) = -N(N - 1)\epsilon/2$, consistent with the simplex solution. In Fig. 7, we have marked with a red line the boundary between the first zone, of fast energy decay, and the second zone of slow decay of energy values.

We tested different functions to fit the data and what provided the most accurate representation of the energy values was to consider a quadratic fit for the first zone of fast energy decay and a linear fit for the second zone of moderate energy decay. Hence, for each value of $N$, we have that the best fit of the global minima energy data as a function of the dimension $D$ can be described by:

$$
E_{GM}(D) = \begin{cases} aD^2 + bD + c & : D \leq \lceil N/2 \rceil \,, \\ mD + n & : D \geq \lceil N/2 \rceil \,, \end{cases}
$$

where the values of $a$, $b$, $c$, $m$ and $n$ can be adjusted for each value of $N$.

Since the values of $a$, $b$, $c$, $m$ and $n$, must be determined for each value of $N$, we look for functions $a = a(N)$, $b = b(N)$, $c = c(N)$, $m = m(N)$, and $n = n(N)$ such that they fit the data of the global-minima energies for $9 \leq N \leq 16$, that can be used to extrapolate to larger values of $N$. We report first the fit of the second zone data in Table 3 for $m$ and $n$ as a function of $N$.

Table 3: Parameters $m$, and $n$ for each $N$ value in the second range of $D$ values (higher dimensional range).

| $N$ | $m$ | $n$ |
|-----|-----|-----|
| 9 | -1.055 | -27.617 |
| 10 | -0.987 | -36.137 |
| 11 | -0.987 | -45.166 |
| 12 | -0.945 | -55.601 |
| 13 | -0.947 | -66.648 |
| 14 | -0.915 | -79.076 |
| 15 | -0.916 | -92.164 |
| 16 | -0.897 | -106.569 |

By observing the data in Table 3 we notice that the values of $n$ are strictly decreasing as $N$ increases. Surprisingly, we notice a different behavior in the values of $m$ depending on whether $N$ is even or odd. It is observed that for two consecutive values of $N$, such that if $N_j$ is even and $N_{j+1}$ is odd, one has that $m(N_j) \simeq m(N_{j+1})$.

Considering the previous paragraph, we tried and analyzed several functions to fit data of Table 3, and the best results were obtained with:

$$m(N) = \frac{-0.773595 \left(2 \lfloor N/2 \rfloor\right)}{2 \lfloor N/2 \rfloor - 2.15249} \ , \tag{10}$$

$$n(N) = -0.499096 N^2 + 1.22355\, N + 1.69286 \ . \tag{11}$$

Figure 8 shows the fits for the function $m(N)$. The data fitted by the function $m(N)$ corresponds to the slopes of the straight lines representing the energies of the global minima of the Lennard-Jones potential in the zone $D \geq \lceil N/2 \rceil$ for each value of $N$. Since for each value of $N$ the energy of the global minimum is decreasing as $D$ increases, we are sure that $m(N) < 0$ for every value of $N$.

Therefore, although other functions may well fit the data of $m(N)$, which as we see in Fig. 8 are increasing, we must make sure that these functions must be negative (and increasing) for all values of $N$, which we have considered in the choice of the function $m(N)$

given by Eq. 10. Moreover, we have come up with a function that accounts for the scaling of the $m$ data in Table 3. At this point we believe it is appropriate to recall that the function $m(N)$ and $n(N)$ are only valid for integer values of $N$. The continuous line added in Fig. 8 is only for illustrative purposes to account for the behavior of the data and corresponds to a modification of the function $m(N)$ given by Eq. 10, where we have replaced $2\lfloor N/2 \rfloor$ for $N$.

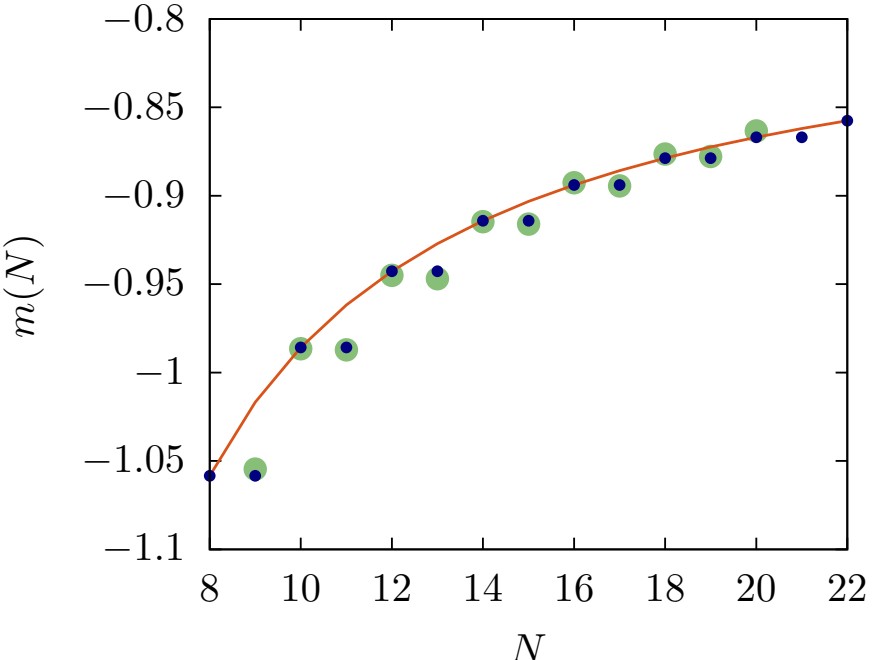

Figure 8: Rational fit $m(N)$. Green dots corresponds to the data of table 3. Black dots corresponds to the modeling of the data throught te function $m(N)$ given by Eq. 10.

Figure 9 shows the fit of the data for $n(N)$. Let us recall that the function $n(N)$ corresponds to the intercept of the straight line representing the energy of the global minima for high dimensions with the vertical axis corresponding to the energy of the global minima (see Fig. 7). Since the slope of such straight lines are negative, the intercept, *i.e.*, $n(N)$ must be greater than $E_{GM}(N, D = N - 1) = -N(N-1)\epsilon/2$, so we have also plotted such a curve with a yellow line as a reference.

Since the number of local minima of the LJ potential for $N$ atoms is considerably smaller in high dimensions $(D \geq \lceil N/2 \rceil)$ compared to lower dimensions $(1 \leq d \leq N - 1)$, we have

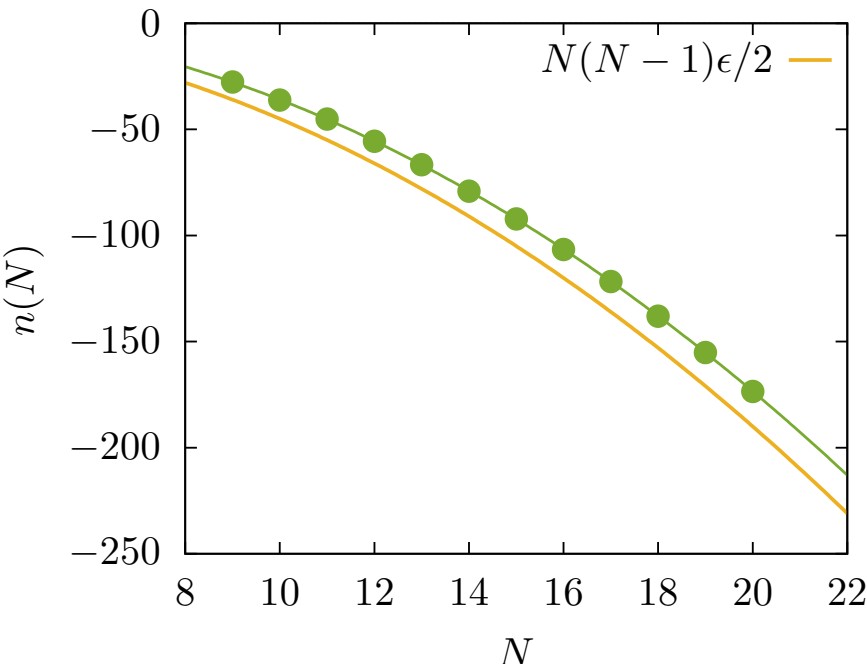

Figure 9: Fit of $n(N)$. Green dots corresponds to data from Table 3. Green line corresponds to the function $n(N)$ given by Eq. 11. Yellow line corresponds to the minumum energy for a cluster of $N$ atoms in dimension $D = N - 1$.

extended the search for local and global minima up to $N = 20$ atoms in this zone and have added the corresponding $m$ and $n$ data in Figs. 8 and 9 for $9 \leq N \leq 20$ to see how our $m(N)$ and $n(N)$ functions behave for values of $N$ beyond $N = 16$.

To fit the data for the first zone in Fig. 7 ($D \leq \lceil N/2 \rceil$), we tried different functions, and the one that best fit the data was a quadratic equation for each value of $N$ defined with the following three points $P_1 = (D = 1, L_1)$, $P_2 = (D = 3, L_2)$, and $P_3 = (D = \rceil N/2 \lceil, L_3)$, where $L_1$ , $L_2$, and $L_3$ correspond to the energies of the global minima in dimensions 1, 3, and $\rceil N/2 \lceil$ respectively, for a given value of $N$.

Since for any value of $N$, we can define the quadratic equation with the aforementioned points, we do not need to calculate each coefficient for each value of $N$, but it suffices to obtain an analytic expression for each coefficient in terms of $P1$, $P2$, and $P3$. Thus, the coefficients of the quadratic function will be given by:

$$a(N) = \frac{-\lceil N/2 \rceil L_1 + \lceil N/2 \rceil L_2 + 3L_1 - L_2 - 2L_3}{-2\lceil N/2 \rceil^2 + 8\lceil N/2 \rceil - 6} \; , \tag{12}$$

$$b(N) = \frac{\lceil N/2 \rceil^2 L_1 - \lceil N/2 \rceil^2 L_2 - 9L_1 + L_2 + 8L_3}{-2\lceil N/2 \rceil^2 + 8\lceil N/2 \rceil - 6} \; , \tag{13}$$

$$c(N) = \frac{\lceil N/2 \rceil^2 (L_2 - 3L_1) + \lceil N/2 \rceil (9L_1 - L_2) - 6L_3}{-2\lceil N/2 \rceil^2 + 8\lceil N/2 \rceil - 6} \; , \tag{14}$$

where $L_1$, and $L_2$ are linear functions of $N$ to fit the data of the global minimum energy in dimension $D = 1$, and $D = 3$ respectively as $N$ increases, and $L_3$ is a quadratic function to fit the data of the global minimum energy in dimension $D = \lfloor N/2 \rfloor$ as $N$ increases. The expressions for $L_1$, $L_2$, and $L_3$ are as follows:

$$L_1 = -1.03473N + 1.07413 \ , \tag{15}$$

$$L_2 = -4.76207N + 18.9534 \ , \tag{16}$$

$$L_3 = -0.500018N^2 + 0.901622N - 0.430931 \ . \tag{17}$$

Figure 10 shows the data for $L_1(N)$, $L_2(N)$ and $L_3(N)$ together with their respective fits given by Eqs. (13-17).

Our model provides reasonable estimates for the ground state energy of Lennard-Jones clusters, particularly for values of $N$ within or near the training range. For instance, at $N = 13$, the predicted total energy deviates by only 3.1% from the known value, with an absolute error of $1.37\,\epsilon$, corresponding to an error of approximately $0.106\,\epsilon$ per atom. For $N = 38$, which lies outside the training range, the deviation increases to 6.85%, with a total error of $11.92\,\epsilon$, or $0.314\,\epsilon$ per atom. In the case of $N = 55$, the discrepancy becomes more pronounced: the predicted energy differs by $36.29\,\epsilon$, yielding an error per atom of approximately $0.660\,\epsilon$. These results highlight the limitations of the model when extrapolating far from the training region, but also suggest that for moderate values of $N$, the estimates remain within a reasonably acceptable margin in Lennard-Jones units. Of course, there are no guarantees, so far, that the minimum energy reported for $N = 38$ and $N = 55$ are indeed the global minimum configuration.

We believe it is important to comment on the strengths and weaknesses of the adjustments made. Since we are trying to make a fit, hopefully for large value of $N$, with only a few and small values of $N$ (let us recall that $9 \leq N \leq 16$) it is possible that more than one function will be able to fit the data properly. The problem with this is that for large values of $N$, our model might not properly estimate the energy of the global minima of the Lennard-Jones potential in case we choose the wrong function to fit the data. Certainly, if someday computational capabilities allow us to collect data for larger values of $N$, especially

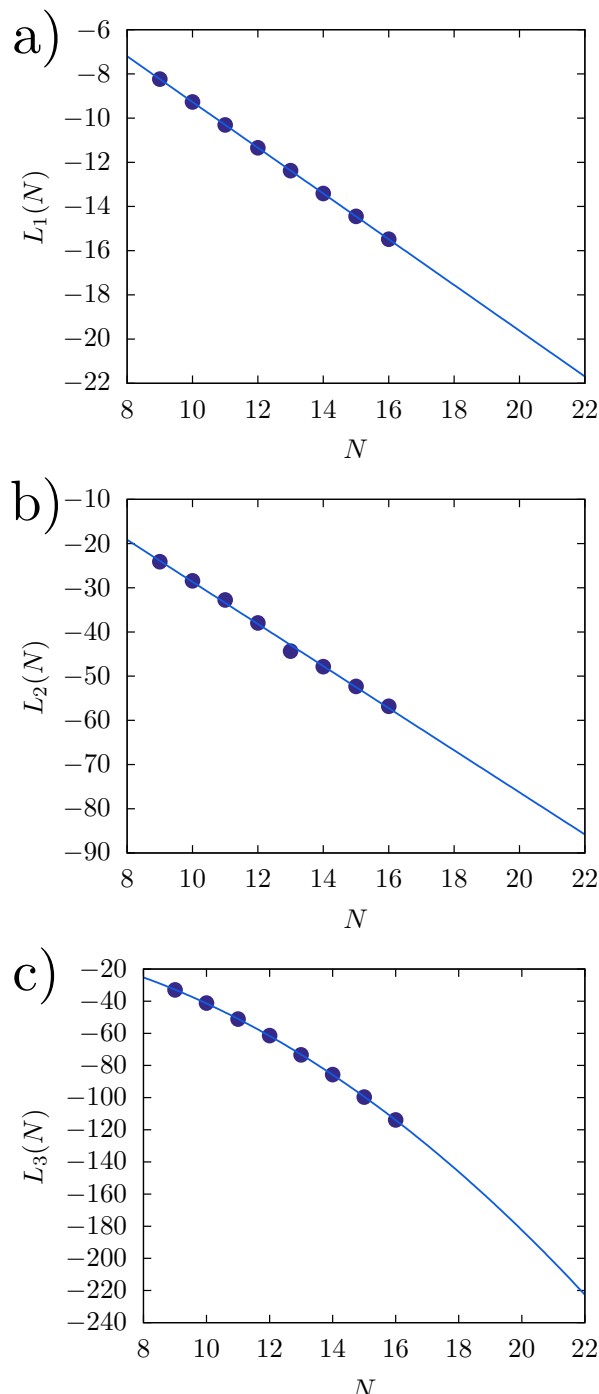

Figure 10: Panels a), b), and c) show the values of $L_1(N)$, $L_(N)$, and $L_3(N)$, respectively, along with their corresponding fits.

at $1 \leq D \leq \lceil N/2 \rceil$, the parameters of the fit can be updated, and as a consequence, estimate more accurate values of both the number of local minima and the energies of the global minima.

On the other hand, regardless of the choice of the proposed fits of the data, it is clear that there are two regimes, the first one for $D \leq \lceil N/2 \rceil$ and the second one for $D \geq \lceil N/2 \rceil$. These two distinct regimes can be explained by examining how dimensionality affects the possible spatial arrangements of particles and, consequently, the system's total potential energy.

In low-dimensional spaces, particles are confined within a limited number of degrees of freedom. This spatial restriction limits the configurations available for minimizing the potential energy. Particles are more likely to be in close proximity, often at distances shorter than the equilibrium separation defined by the Lennard-Jones potential, resulting in increased repulsive interactions and a higher total energy. As the dimensionality increases from a low value, the system gains additional degrees of freedom. This allows particles to rearrange more effectively, moving towards configurations where they are at or near the equilibrium separation. The increased ability to avoid unfavorable close contacts leads to a substantial decrease in the system's potential energy. The proposed quadratic decrease in energy with increasing dimension in this regime suggests that each additional dimension provides a new independent direction for particles to separate and minimize repulsion. The energy reduction is more pronounced at lower dimensions because the relative increase in available configurations is significant with each added dimension.

In higher-dimensional spaces, the particles have already exploited most of the available degrees of freedom to minimize their mutual repulsion. The spatial configurations have approached an optimal arrangement where particles are at or near equilibrium separations. Further increases in dimensionality provide diminishing returns in terms of energy reduction. The particles can only make minor adjustments to their positions, leading to marginal decreases in the total potential energy. The system is effectively in a state where additional dimensions offer little advantage for energy minimization. The linear trend observed in this

regime indicates that each additional dimension contributes a relatively constant but small decrease in energy. This suggests that the particles are making fine adjustments rather than significant rearrangements, resulting in a slower rate of energy reduction.

# 4  Conclusions

In this study, we have developed functions to estimate the number of local minima of the Lennard-Jones potential and the energies of the global minima for any number of atoms $N$ and dimensions $D$ between 1 and $N - 1$. Based on empirical data from atomic structures where $9 \leq N \leq 16$, and $1 \leq D \leq N - 1$, our proposed functions have demonstrated accuracy and general applicability within these ranges. Moreover, they can be extrapolated to estimate properties in systems with values of $N$ and $D$ beyond those studied.

As for the number of local minima, we have proposed a function that estimates the diversity of stable structures in terms of the number of atoms and dimension. The proposed function establishes in a natural way, that the maximum structural diversity is found in dimension $D = 3$, independent of the number of atoms, which can be explained in the framework of a balance between the degrees of freedom and the atomic interactions of the system.

We have also explored the use of machine learning techniques to predict the number of local minima of the Lennard-Jones potential. In particular, we train neural networks using data obtained for relatively large values of the dimension $D$, which allows the models to capture structural patterns in the energy landscape in high-dimensional regimes. Remarkably, these networks are able to extrapolate with good accuracy the number of local minima in lower dimensions, including values close to $D = 3$, which are of particular physical relevance. This approach provides a complementary tool to traditional analytical and numerical methods, and suggests that there are deep regularities in the configuration space structure that can be effectively captured through machine learning models, and that this structure can be

uncovered if we consider its behavior in different dimension.

On the other hand, the energy behavior across different dimensional regimes can be attributed to the interplay between geometric constraints and the availability of degrees of freedom for particle rearrangement. In low dimensions, particles are highly constrained, and increasing the dimensionality significantly enhances their ability to rearrange and reduce repulsive interactions. This leads to a rapid, quadratic decrease in energy as the dimensionality increases. In high dimensions, particles have already optimized their positions to minimize energy. Additional dimensions allow only slight adjustments, resulting in a slower, linear decrease in energy. This analysis highlights how dimensionality plays a crucial role in the energetic optimization of particle systems and provides insight into the fundamental nature of intermolecular interactions across different spatial dimensions.

The analytical methods introduced should offer reliable estimations that traditionally require intensive computational simulations. This advancement significantly reduces computational costs and accelerates the research process by facilitating the identification of the most stable configurations without exhaustive calculations.

Our results enrich the theoretical understanding of the Lennard-Jones potential's behavior across different dimensions and system sizes, providing new insights into interatomic interactions. By addressing dimensions $D$ between 1 and $N-1$, we have expanded the understanding of systems in higher-dimensional spaces, which is relevant in fields such as theoretical physics and quantum chemistry.

We believe the functions we have developed serve as valuable tools for the scientific community in molecular modeling and simulation, particularly in predicting energetic and structural properties. They hold significant relevance for nanotechnology and materials science, where precise estimations of local minima and associated energies are crucial for the design and synthesis of new nanoscale materials.

Also we believe that the strategy described in the manuscript can be used to study other potentials, being pair or multibody, where a multidimensional analysis can provide useful

additional information.

In summary, the functions proposed not only offer practical solutions for estimating fundamental characteristics of the Lennard-Jones potential, which is a highly used potential in different molecular dynamic simulation codes, but also lay the groundwork for future research. They can be utilized in more complex systems or adapted for similar methods in other interatomic potentials, thereby contributing to the advancement of knowledge in atomic and molecular system studies.

# Acknowledgement

This work was funded by the National Agency for Research and Development (ANID) under Fondecyt award numbers 1240697 (JAV) and 1240655 (JR).

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
