# Peer review of "Multidimensional Modeling of the Number of Local Minimum-Energy Structures and the Energy of Putative Global Minima"

_SciPost Physics Core_

## Round 1 · Referee Report · Anonymous (Referee 1) · 2025-11-23

Report

In this manuscript, the authors develop a modeling framework for Lennard-Jones clusters that interpolates and extrapolates both the number of local minimum-energy structures and the energies of global minima as functions of atom number N and spatial dimension D. The analysis shows an interesting picture. Thus, the structural diversity, quantified via the number of local minima, is maximized around D=3, and the global minimum energies exhibit two distinct regimes separated near D=N/2. The work is technically solid, the modeling choices are well motivated, and the numerical evidence is presented clearly. The physical interpretation of the dimensional trends is convincing and, in my view, of broad interest to the chemists and physics communities. I recommend this manuscript for publication in its present form.

Requested changes

Figure 5: in the caption replace "The yellow line " with "The yellow region" or provide a contrast line for maximum diversity path.
Page 18: in "We may wonder why the above multidimensional discussion is import. " replace "import" with "important".

Recommendation

Publish (surpasses expectations and criteria for this Journal; among top 10%)

---

## Round 1 · Referee Report · Anonymous (Referee 2) · 2025-12-5

Report

I have several observations regarding this paper, concerning both the form and the content.

In what follows I will list the main observations:
\begin{itemize}
\item The expression "scarce--and--limited" that appears in the abstract (and later on in the text) is puzzling: what exactly do the authors want to say?

\item Still in the abstract, where the authors mention the "maximum diversity of stable structures": it is not clear to me whether the authors mean that there are more local minima for $D=3$, or if the configurations of these minima also differ more between them. In the first case, the sentence should be written more clearly, while in the second case, the specific meaning of "diversity" should be stated.

\item The first line of the Introduction contain a "dimentional", whereas at the middle of the same page one reads "\dots allow us to rationalize an finally understood \dots ", which also is incorrect. Also at page 4, "Another challenge \dots challenging \dots". Again at pag 13, in the caption of the figure, "dimensionn". Pag. 18, "\dots is import".

\item Pag. 3, where it say "Remarkably, 2024 marks the 100th anniversary \dots": we are in 2025 and almost in 2026, so this sentence is incorrect.

\item Pag. 4, the sentence "But even as $N$ increases the certainty \dots": I find this sentence obscure.

\item Pag. 4, the sentence "To date, there are only estimates that as $N$ increases, the number of local minima grows exponentially. Such estimates are based on the number of local minima found for $N\leq 15$ and, of course, dimension $D=3$".

The authors are studying the Lennard--Jones potential, however, the exponential growth of the number of local minima is observed in a large class of problems involving
either larger range (for instance Coulomb) and shorter range (packing) potentials. Because of the exponential growth of the number of local minima with $N$, finding the global minimum becomes particularly challenging when $N$ is large enough. The authors should include this discussion, with the appropriate references.

\item Pag. 5, "All local minima obtained were checked using the eigenvalues of the Hessian matrix, and to ensure that each local minimum is unique \dots we use the ordered eigenvalues of the Coulomb matrix".

I think that the authors need to explain in more detail this last procedure and why it allows to determine whether two configurations are different or not.
I want also to point out that, since the configurations are obtained numerically, the authors should provide more details both on the numerical calculation performed and
on the accuracy of the numerical results. What are the typical sizes of the gradients of the configurations that have converged to a local minimum? Are there situations in which the finite precision of the numerical results will make the comparison inconclusive (maybe it is unlikely for $N \leq 15$ but it could become an issue at larger $N$)?
And, finally, is the accuracy of the numerical results comparable for different $D$?

\item Pag.7, "For the specific case $D=3$, when $N \leq 13$ our method was able to find all the local minima reported in previous works[14]".

First of all Ref.14 is a single paper, so "work" should be more appropriate; secondly, this reference is by some of the authors in this paper, so the sentence would make sense if two different methods/programs were used in this searches. Even assuming that this is the case, it is not clear whether if the authors are just saying that the number of configurations is the same or if they have compared all the new configurations with the old ones, finding that they are equivalent.

\item At pag.11 the authors mention that a reduction in precision should be expected as $N$ grows, but then at pag. 13, they consider $N=147$ and in table 2 of pag.16
the report estimates for $N=38,55,147$. The values are compared with those "previously reported" (where? there is reference here). Does it really make any sense to use a fit obtained from data for $N \leq 16$ to $N=147$? To me, the only message that this table conveys is that the number of minima is huge, which itself is not a big surprise.

\item Pag. 14, "The yellow color \dots where the diversity is maximum \dots": as before, I am not sure of the meaning of "diversity".

\item Pag. 16, "Lower packing density implies fewer way to arrange atoms in dense and energetically favorable configurations". This is presented as a self--evident fact, but
I don't find obvious. The authors have to be more precise.

\item Pag. 23: the black dots mentioned in the caption of the figure to me look blue.

\end{itemize}

In addition to the above, there are several objections I feel to make:
\begin{itemize}
\item First of all, very little detail is given on the numerics and on the algorithms implemented in the program: how does the program looks for new configurations? how many trials are being performed? how precise is the energy of the configurations obtained?

\item I find dangerous to extrapolate formulas obtained over a rather small range of values to much larger values; it would make sense, on the other hand to conduct an exploration for $N=17$ and see whether it confirms what expected on the grounds of the fits;

\item the discussion of the authors is uniquely centered on the number of minima: since they have calculated a large population of configurations, it would make sense to try to classify them (for example, they can look at the coordination numbers of the particles) and compare them;

\item Given the complexity of the problem, extra care should be put in insuring that the exploration for $D >3$ is not missing configurations: since the authors have at their disposal the configurations for $D= 3$ (and $D=2$) they should observe that these configurations, when "transplanted" in a dimension $D'>D$ become stationary points.
By perturbing these configurations, and then minimizing, one would land in a local minimum for $D'>3$. If the Hessian has several negative eigenvalues, these perturbations may possibly produce several -- potentially different -- configurations. Of course, if the number of local minima for $D'>3$ has to be smaller than for $D=3$, one should see that different minima for $D=3$ collapse into the same minimum for $D'>3$. This experiment should be rather easy to perform. Similarly, configurations
for $D'>3$ could be reduced to configurations for $D=3$ by adding an external confining potential in one or more directions (depending on $D'$) and progressively squeezing them.

\item The configurations obtained numerically should be made available to the readers, for example using Zenodo.

\vspace{1cm}

The paper cannot be accepted in the present form and I believe that a substantial amount of work is needed.

Recommendation

Reject

  • validity: ok
  • significance: ok
  • originality: ok
  • clarity: low
  • formatting: below threshold
  • grammar: below threshold

---

## Editorial Decision

awaiting_resubmission